An update on anatomy and function of the teleost olfactory system

Olivares Jesús 1 2
Schmachtenberg Oliver oliver.schmachtenberg@uv.cl 1
1 Centro Interdisciplinario de Neurociencia de Valparaíso (CINV), Universidad de Valparaíso , Valparaíso , Chile
2 Universidad de Valparaíso, PhD Program in Neuroscience , Valparaíso , Chile
Esteban María Ángeles
Electronic publication date: 2019 Sep 27
Publication date: 2019
Volume: 7
Electronic Location ID: e7808
Received 2019 Jul 23; Accepted 2019 Sep 1
Copyright: ©2019 Olivares and Schmachtenberg
Copyright year: 2019
Copyright holder: Olivares and Schmachtenberg
License: This is an open access article distributed under the terms of the Creative Commons Attribution License, which permits unrestricted use, distribution, reproduction and adaptation in any medium and for any purpose provided that it is properly attributed. For attribution, the original author(s), title, publication source (PeerJ) and either DOI or URL of the article must be cited.
License URL: https://creativecommons.org/licenses/by/4.0/

Keywords: Teleost, Fish, Olfaction, Sense of smell, Olfactory bulb, Telencephalon, Olfactory receptor neurons, Odor coding, Olfactory transduction

Funding: Chilean government through FONDECYT 1171228 Millennium Institute CINV P09-022-F This work was supported by the Chilean government through FONDECYT grant No. 1171228, a CONICYT PhD scholarship, and the Millennium Institute CINV (P09-022-F). The funders had no role in study design, data collection and analysis, decision to publish, or preparation of the manuscript.

==============================
About half of all extant vertebrates are teleost fishes. Although our knowledge about anatomy and function of their olfactory systems still lags behind that of mammals, recent advances in cellular and molecular biology have provided us with a wealth of novel information about the sense of smell in this important animal group. Its paired olfactory organs contain up to five types of olfactory receptor neurons expressing OR, TAAR, VR1- and VR2-class odorant receptors associated with individual transduction machineries. The different types of receptor neurons are preferentially tuned towards particular classes of odorants, that are associated with specific behaviors, such as feeding, mating or migration. We discuss the connections of the receptor neurons in the olfactory bulb, the differences in bulbar circuitry compared to mammals, and the characteristics of second order projections to telencephalic olfactory areas, considering the everted ontogeny of the teleost telencephalon. The review concludes with a brief overview of current theories about odor coding and the prominent neural oscillations observed in the teleost olfactory system.

Introduction

The sense of smell plays an important role in fishes, mediating behaviors and physiological responses related to food search and feeding, social interaction, mating, detection of predators and contamination, or migration and search for spawning sites (Sorensen & Caprio, 1998; Bone & Moore, 2008). As opposed to higher vertebrates (amphibians, reptiles and mammals), which usually have access to a limited degree of light even in moonless nights, fish encounter situations of total darkness easily in turbid or deeper waters and underwater caves, where their olfactory sense may provide a degree of orientation. Olfactory information has long been known to be important for the homing of migrating salmonids and other fishes (Dittman & Quinn, 1996; Bett & Hinch, 2016). There is evidence for salmonids that some amino acids dissolved in the water of native rivers could mediate migratory conducts (Yamamoto, Hino & Ueda, 2010; Yamamoto, Shibata & Ueda, 2013). However, in general, the identity of the odors mediating orientation behavior in teleosts remains a topic of active investigation. On the other hand, amino acids are considered the most relevant odorants indicating the presence of a food source to teleosts, given the behavioral responses they generate (Valentinčič, Lamb & Caprio, 1999; Whitlock, 2006; Calvo-Ochoa & Byrd-Jacobs, 2019), and the fact that teleosts are generally sensitive to most amino acids, with electrophysiological screening thresholds ranging from micromolar to nanomolar concentrations (10−6–10−8 M) (Ivanova & Caprio, 1993; Nikonov & Caprio, 2007; Bazáes, Olivares & Schmachtenberg, 2013; Nikonov et al., 2017; Sato & Sorensen, 2018). Nucleic acids and polyamines are also related to food odor, with evidence that several nucleotides, including ATP and adenosine, are able to evoke olfactory responses in channel catfish (Ictalurus punctatus), zebrafish (Danio rerio) and goldfish (Carassius auratus) (Hansen et al., 2003; Rolen et al., 2003; Wakisaka et al., 2017).

Sensitivity thresholds for different types of odorants depend evidently on the species, but also on the type of assay, behavioral versus electrophysiological, and for the latter, whether individual neurons or bulk responses are recorded from the olfactory epithelium, bulb or telencephalon. Studies using electroolfactogram (EOG) recordings have indicated that fish are sensitive to very low concentrations of certain odorants, presenting responses to some amino acids at thresholds in the order of 10−9 M, and below 10−11 M for certain bile salts, steroid hormones and prostaglandins (Reutter & Kapoor, 2005; Bone & Moore, 2008). Bile salts are good candidates for social odors, as they are constantly released by fishes (Buchinger, Li & Johnson, 2014). However, controversy persists as to what degree mixtures of bile salts are species-specific and may therefore be considered pheromonal odors (Sorensen & Stacey, 2004; Giaquinto & Hara, 2008). Recently, using EOG recordings in the rainbow trout, responses to lactic acid, pyruvic acid and four B vitamins with thresholds in the nanomolar range been reported for the first time in rainbow trout, adding these compounds to the growing list of food odors. Lactate and pyruvate are important components of cellular metabolism, while B vitamins, which are synthesized by phytoplankton and algae, are essential for teleost development, which might explain their sensitive detection (Valdés et al., 2015).

Some odorants trigger an instinctive avoidance response. For example, an imminent risk of predation can be detected by fish through the presence of passively released molecules when a conspecific is attacked and injured, which prevail in the environment for much longer than visual or mechanical signals. The skin of teleosts presents a type of cells called club cells that release a combination of chemical warning signals in the context of predation or parasitism, generating changes in the behavior of rainbow trout and other teleost species. These substances include glycosaminoglycans of chondroitin, an important component of the skin’s mucosa (Poulin, Marcogliese & McLaughlin, 1999; Speedie & Gerlai, 2008; Wisenden et al., 2009; Mathuru et al., 2012).

Here, we provide an update on anatomy and function of the teleost olfactory system, including the important advances achieved during the last years. The review is intended to complement prior major revisions of the relevant literature on fish olfaction, covering peripheral coding and olfactory bulb circuitry (Satou, 1990; Laberge & Hara, 2001), the olfactory gene repertoire (Korsching, 2009), the variety of olfactory receptor neurons and their respective connections (Hansen et al., 2005; Hamdani & Døving, 2007; Bazáes, Olivares & Schmachtenberg, 2013; Yoshihara, 2014) and the generation of olfactory behaviors (Kermen et al., 2013; Yoshihara, 2014).

Survey methodology

Initially, a PubMed search was conducted for the terms “teleost” and “olfactory”, yielding 304 results. Twenty-two notable and highly cited articles on vertebrate olfaction were also included in the general data base. Four books were additionally considered: “Fish Chemosenses” (Reutter & Kapoor, 2005); “Biology of Fishes” (Bone & Moore, 2008); “Comparative Vertebrate Neuroanatomy” (Butler & Hodos, 2005) and “Sensory Systems Neuroscience” (Hara & Zielinski, 2007). A total of 116 references was finally selected and included in the review. It should be noted that while the field of teleost olfaction is small compared to other biomedical areas, it is still too large to be comprehensively covered in one review, and no attempt was made in that direction. In particular, an inter-species comparison was beyond the focus of this work, since the large number of teleosts and their huge variety would render any such attempt hopeless. Instead, we focused on the most popular model species: Channel catfish, rainbow trout, goldfish and zebrafish.

Olfactory epithelium and nerve

The nasal cavities of teleost fishes, including salmonids such as rainbow trout (Oncorhynchus mykiss), are located on the dorsal anterior face of the head (Fig. 1A, inset). Each cavity has two openings, allowing entry and outflow of water, propelled by the movement of the fish, muscular pumping actions, ciliary beating of supporting cells, and water currents (Doving et al., 1977). Unlike higher vertebrates, there is no connection between the olfactory and the respiratory systems, and respiratory sniffing and its implications for odor detection are absent, although some fishes may use muscular contractions for active olfactory sampling (Doving et al., 1977; Nevitt, 1991). Teleost fishes lack a separate vomeronasal organ, and the sole olfactory epithelium is located on the floor of the nasal cavity, in most species arranged in the form of sheets or lamellae, which radiate from a central ridge or raphe and give rise to an olfactory rosette (Hansen & Zielinski, 2005) (Fig. 1A). The number and size of the lamellae increase throughout development of the teleost, but remain relatively constant after the specimen reaches maturity. The lamellae are composed of two pseudostratified epithelial layers that wrap a thin sheet of stroma. Sensory and non-sensory cells are irregularly interspersed within the epithelia, as are mucus-producing goblet cells (Bazáes, Olivares & Schmachtenberg, 2013) (Fig. 1B).

Figure 1 Rainbow trout (Oncorhynchus mykiss) as model system for teleost olfactory research.

(A) Isolated olfactory rosette of a juvenile specimen. Inset: Exposed olfactory organs of O. mykiss. (B) Toluidine blue-stained cryosection of the olfactory lamellae from O. mykiss. Inset: Ultrastructural detail of the dendritic endings of a microvillous (m) and a ciliated (c) ORN. (C) Horizontal cryosection through the olfactory bulbs, stained with toluidine blue. (D) Transverse section through the sensory neuroepithelium, immunohistochemically labeled for Gao (red) to mark microvillous ORNs, and for PCNA (green) to label basal cells. Modified from Bazáes et al., 2013. (E) Synchronous responses to a mixture of amino acids, recorded from the olfactory epithelium (EOG), olfactory bulb (OB), and telencephalic areas Vv and Dp in a live anesthetized specimen of O. mykiss. Note prominent field potential oscillations during the odor response. From Olivares (2019), PhD thesis, Universidad de Valparaiso. (F) Top: Schematic drawing of the rainbow trout brain. The telencephalic olfactory areas Dp and Vv are indicated in their respective sections. Below, ORN types, olfactory bulb circuitry and tracts to the telencephalon. ORN, Olfactory Receptor Neuron; OB, Olfactory bulb; Tel, Telencephalon; OT, Optic tectum; Cb, Cerebellum; R, Ruffed cell; MC, Mitral Cell; Gr, Granule Cell; CF, Centrifugal fiber; Vv, Ventral nucleus of ventral telencephalon; Dp, Dorsal posterior area; EOG, electroolfactogram. Scale bars: A, 1 mm, B, 300 µm; C, 500 µm, D, 100 µm.

The olfactory epithelium consists of three main types of cells: Olfactory receptor neurons (ORNs), supporting or sustentacular cells, and basal cells, which are capable to divide and regenerate the epithelium after injury (Iqbal & Byrd-Jacobs, 2010) (Fig. 1D). In teleosts, there are four established types of ORNs intermingled in the olfactory epithelium: Ciliated and microvillous receptor neurons, olfactory crypt cells and Kappe cells, the latter discovered in zebrafish (Hansen et al., 2003; Bazáes, Olivares & Schmachtenberg, 2013; Ahuja et al., 2015). A putative fifth type of ORN, termed “pear-shaped”, was also recently described in zebrafish, and shown to function as an adenosine sensor (Wakisaka et al., 2017). Olfactory cilia and microvilli contain the olfactory receptor proteins that are exposed to chemical stimuli in the water. Ciliated and microvillous ORNs are bipolar cells, with an apical cylindrical dendrite that extends to the surface of the lamellae (Fig. 1B, inset) and an axon that projects towards the submucosa forming the fascicles of the olfactory nerve. These cell types are morphologically and molecularly similar to the ciliated and microvillous ORNs of the main olfactory and vomeronasal organs of higher vertebrates (Farbman, 2000; Hansen et al., 2003). Olfactory crypt cells, which are located in more apical regions of the epithelium, have an oval shape with cilia and microvilli enclosed in an apical invagination, and are completely enveloped by one or two supporting cells of glial characteristics (Hansen & Finger, 2000; Schmachtenberg, 2006; Bazaes & Schmachtenberg, 2012). The Kappe cells, found in the most apical epithelial positions, have a similar shape as crypt cells, but are slightly more slender and do not present cilia (Ahuja et al., 2015). Finally, little is known about the recently described pear-shaped zebrafish ORN, which is morphologically similar to crypt and Kappe cells, but displays different molecular signatures, such as expression of olfactory marker protein (OMP) (Wakisaka et al., 2017). These latter two cell types have hitherto not been described in teleosts other than zebrafish. If they do indeed qualify to be recognized as separate ORN types remains to be confirmed.

Olfactory bulb and tract

The axonal projections of vertebrate ORNs traverse the skull in olfactory nerve fascicles that enter the olfactory bulbs and converge in spherical neuropil complexes termed glomeruli (Mombaerts et al., 1996). The teleost olfactory bulb has several similarities and some differences with respect to the olfactory bulb of other vertebrates (Satou, 1990; Kermen et al., 2013; Nagayama, Homma & Imamura, 2014; Calvo-Ochoa & Byrd-Jacobs, 2019) (Fig. 1C). Recently, an entire olfactory bulb of zebrafish larvae has been reconstructed from serial section electron microscopy, including the skeletons of over 1,000 neurons, 75% of which were identified as mitral cells (Wanner et al., 2016; Wanner, Genoud & Friedrich, 2016). In the teleost olfactory bulb, four layers can be identified from the periphery towards the center: Olfactory nerve layer, glomerular layer, mitral cell layer and granular layer (Satou, 1990). The outermost layer of this structure corresponds to the layer formed by the axonal endings of the ORNs, which are distributed profusely over the surface of the bulb before traversing deeper into the glomeruli of the glomerular layer, where they form glutamatergic synapses with the dendrites of the mitral cells, whose cell bodies are located in the subjacent layer (Fujita, Satou & Ueda, 1988). The glomerular layer of teleosts also contains nerve endings from higher telencephalic centers and projections of granule cells, GABAergic cells whose cell bodies are located in the layer that bears the same name and with which the mitral cells share dendrodendritic synapses (Satou, 1990). As opposed to mammals, mitral cells can innervate multiple glomeruli in teleost fish, but not all of them do (Laberge & Hara, 2001; Fuller, Yettaw & Byrd, 2006). Bulbar glomeruli are surrounded by short axon cells and periglomerular cells, which participate in this first stage of olfactory processing, together with neuromodulatory efferent projections from the telencephalon making synapses with granular cells in the granular layer of the bulb (Nieuwenhuys, Ten Donkelaar & Nicholson, 1998) (Fig. 1F).

The tufted cells of mammals are either absent or indistinguishable from mitral cells in the teleost olfactory bulb (Edwards & Michel, 2002). However, a second type of projection neuron termed ruffed cells has been observed in fish that is not present in mammals (Kosaka & Hama, 1979; Kosaka & Hama, 1981). The ruffed cells, which project to the telencephalon together with the mitral cells, also present dendrites in the glomerular layer but only appear to make synapses with granule cells (Kosaka & Hama, 1979; Satou, 1990). These cells are thought to play an important role in olfactory processing and in the generation of prominent olfactory oscillations in the teleost bulb, since their response is of opposite polarity compared to that of mitral cells (Zippel, 1998) (Fig. 1E). The axons of the mitral cells give rise to the olfactory tracts that carry the information to other relay centers in the telencephalon (Nieuwenhuys, Ten Donkelaar & Nicholson, 1998) (Fig. 1F). Studies using tracers, electrophysiology, genetics and behavior in rainbow trout (Oncorhynchus mykiss), zebrafish (Danio rerio), goldfish (Carassius auratus), crucian carp (Carassius carassius) and other teleosts have allowed to determine the presence of two subdivisions of the olfactory tract: the lateral olfactory tract and the medial olfactory tract (Folgueira, Anadón & Yáñez, 2004a; Hamdani & Døving, 2007; Miyasaka et al., 2009; Miyasaka et al., 2014). These can be further subdivided into fiber bundles which conserve the chemotopic organization of the olfactory bulb to some degree, thus maintaining the odotopically organized processing of different types of odors (Finger, 1975; Satou, 1990). In crucian carp, ablation studies of the different olfactory tracts that project to various telencephalic regions indicated that the medial bundle of the median olfactory tract mediates escape behavior in response to odors considered as alarm signals (Hamdani et al., 2000), while the lateral bundle of the median olfactory tract relays odor information related to reproductive behavior (Weltzien et al., 2003). Finally, the lateral olfactory tract has been reported to convey information about odorants related to feeding behaviors in this species (Hamdani, Alexander & Døving, 2001).

The odor transduction process

In the ciliary and microvillous membranes of vertebrate ORNs, odorant receptor proteins belonging to the OR superfamily of G protein-coupled receptors are expressed (Buck & Axel, 1991). In addition, a subset of receptor neurons express another class of odorant receptor proteins, termed trace-amine associated receptors (TAARs) (Liberles & Buck, 2006). A third class of receptor proteins, the vomeronasal receptors (VRs), is expressed within the vomeronasal organ of many higher vertebrates, and in the olfactory epithelium of fishes, where it may serve to detect conspecific odors and pheromones (Silva & Antunes, 2016). It was demonstrated in several teleost species that the different types of olfactory receptor proteins are expressed differentially in the specific ORN types, which in turn present different transduction processes associated with diverse types of G-proteins (Hansen et al., 2003; Hansen, Anderson & Finger, 2004). While ciliated ORNs generally express receptor proteins of the OR and TAAR types, which are associated with Gαolf proteins, microvillous receptor neurons present V1R-type receptors associated with Gαi, and V2R receptors coupled to the G-proteins Gαo (Fig. 1D), Gαq, Gαq∕11 and Gαi−3. Crypt cells express V1R type receptors coupled to Gαo, Gαq∕11 and Gαi1b, while Kappe cells, although their class of odorant receptor proteins remains unknown, are thought to be associated with Gαo proteins (Hansen et al., 2003; Hansen, Anderson & Finger, 2004; Hansen & Zielinski, 2005; Bazáes, Olivares & Schmachtenberg, 2013; Ahuja et al., 2015).

The olfactory receptor gene repertoire has been analyzed in several teleost species, revealing a significant degree of variation and diversity, with 189 functional genes and 31 pseudogenes detected in zebrafish, the highest number in teleosts counted to date (Korsching, 2009; Yoshihara, 2014). In the case of salmonid fishes, a total of 60 functional genes and 51 pseudogenes have been found in Salmo salar, which are distributed among the different families of olfactory receptors in such a way that 24 functional OR-type receptors have been found together with 24 pseudogenes, seven genes of functional V1R-type receptors and one pseudogene, and 29 genes of V2R-ype receptors with 26 pseudogenes (Johnstone et al., 2012). More recently, the discovery of 27 putative functional genes and 25 putative pseudogenes of the TAAR family has been added to this list (Tessarolo et al., 2014).

The conformational change that the receptor undergoes due to the binding of an odorant induces the activation of the respective G proteins, which dissociate their alpha subunit. In ciliated ORNs of teleosts, as in those of higher vertebrates (Buck & Axel, 1991), the alpha subunit of the Gαolf protein triggers the activation of the enzyme adenylate cyclase, generating an increase in cyclic AMP (cAMP) that functions as a second messenger (Hansen et al., 2003; Schmachtenberg & Bacigalupo, 2004). cAMP causes the activation of cationic cyclic nucleotide-gated channels (CNGCs), allowing the influx of sodium and to a lesser extent of calcium, which depolarizes the cell membrane directly and by an amplifying effect, through the activation of nearby calcium-dependent chloride channels (Kurahashi & Yau, 1994; Reisert & Reingruber, 2019). In addition, calcium contributes to the termination of the response through an adaptive reduction of the sensitivity of CNGCs to cyclic nucleotides by means of calcium-calmodulin activation, until intracellular calcium levels return to pre-stimulation levels mainly by action of the sodium-calcium exchanger (Matthews & Reisert, 2003). In O. mykiss, the function of calcium-activated chloride channels (termed either Anoctamin 2 or TMEM16B) has been demonstrated directly with patch clamp recordings of ORNs and by electroolfactogram (EOG) recordings of the olfactory organs, indicating that the olfactory transduction process operates in the same way in freshwater fish as in higher vertebrates (Sato & Suzuki, 2000). In seawater, the large extracellular sodium chloride concentration would make the amplificatory outward chloride currents both unnecessary and impossible (Osorio & Schmachtenberg, 2013), however, the absence of Anoctamin 2 or TMEM16B channels from ORNs of marine teleosts remains to be shown. The transduction pathways of the other teleost ORN classes are less well understood, and have been reviewed elsewhere (Bazáes, Olivares & Schmachtenberg, 2013).

The teleost telencephalon

In the brain, sensory input is analyzed and compared with available memories, integrated between different sensory modalities; new memories are formed and behavioral responses may be initiated. Between the main groups of vertebrates, there is a marked difference in the development of the brain, which makes comparative neuroanatomical studies between species difficult. While in vertebrates including lampreys, cartilaginous fish, amphibians and amniotes, the telencephalon develops from the neural tube through a process called evagination, in teleost fishes a process called eversion is generated, in which structures of the brain are distributed differently (Butler, 2000; Butler & Hodos, 2005; Folgueira et al., 2012). Evagination is produced by the expansion of the lumen of the neural tube, which gives rise to the lateral ventricles inside the telencephalon. The eversion process, however, generates a telencephalon without cavities, developing a ventricular surface located dorsally (Nieuwenhuys, 2011). Although the sections of the neural tube that give rise to the different functional cortical regions of species generating the evagination process are homologous to those found in teleost fishes, once the eversion process is completed during embryonic development, the functional regions, including those that process olfactory information, present a disposition that differs greatly from the other groups of vertebrates (Butler & Hodos, 2005; Ito & Yamamoto, 2009; Folgueira et al., 2012). A distinctive feature of the mammalian telencephalon is the presence of a neocortex or six-layer pallium, which is considered responsible for superior cognitive functions. The pallium of the teleost fishes lacks a six-layer cortex, and even laminar structures. Instead, the teleost telencephalon contains zones and nuclei that are neuroanatomically and functionally analogous to the pallial and subpallial structures of evaginating vertebrates (Ito & Yamamoto, 2009).

In the teleost telencephalon, two regions defined by topographic criteria can be recognized: The dorsal area (Area Dorsalis Telencephali), and the ventral area (Area Ventralis Telencephali) considered homologous to the pallium and the subpallium, respectively, of the remainder of vertebrates. Both telencephalic regions were shown to receive bulbar projections in zebrafish (Miyasaka et al., 2014). These telencephalic areas have been subdivided into zones and nuclei by various authors, among which the ventral area of the telencephalon can be distinguished into a central (Vc), ventral (Vv), dorsal (Vd), lateral (Vl); supracomissural (Vs) and paracomissural (Vp) nucleus, while the dorsal area can be subdivided into a central (Dc), dorsal (Dd), lateral (Dl), medial (Dm) and posterior (Dp) zone (Nieuwenhuys, Ten Donkelaar & Nicholson, 1998; Wullimann & Rink, 2002; Folgueira, Anadón & Yáñez, 2004b; Biechl et al., 2017) (Fig. 1F). It should be noted however, that in rainbow trout, the dorsal and the dorsolateral region of the dorsal area of the telencephalon (Dd + Dl −d), have been proposed to form a homogeneous region according to cytoarchitectonic and neurochemical criteria (Folgueira, Anadón & Yáñez, 2004b).

The differences in the development of the teleost telencephalon make it difficult to identify areas that are functionally homologous to the telencephalon of other vertebrates. Studies analyzing neurochemistry, connectivity, gene expression and development in several species of teleosts suggest that the nuclei located dorsally within the Area Ventralis, Vc and Vd, represent the striatum of other vertebrates. Also, the Vs represents subpallial tonsillar regions, and Vv and Vl, located ventrally in the telencephalon, could represent the septum, with Vv being a homologue of the lateral septum, the nucleus accumbens and the substantia innominata, which are related to the reward system. In the dorsal area of the telencephalon, Dc represents the pallium, the most medial part of Dm is thought to be analogous to portions of the amygdala, and Dl is likely homologous to the hippocampus.

Studies in different species of teleosts have established the pallial regions and the subpallial nuclei that receive projections from the olfactory bulb: Dp, Dl, Vv, Vs, Vp, Vc, Vl, among which the Vv region stands out related to the reward systems, and Dp related to the identity of odorants. Dp is the main target of the secondary olfactory fibers (Fig. 1F), comparable to the primary olfactory cortex or piriform cortex of higher vertebrates (Wullimann, Rupp & Reichert, 1996; Nieuwenhuys, Ten Donkelaar & Nicholson, 1998; Cerdá-Reverter, Zanuy & Muñoz Cueto, 2001; Wullimann & Rink, 2002; Folgueira, Anadón & Yáñez, 2004a; Folgueira, Anadón & Yáñez, 2004b; Butler & Hodos, 2005; Mueller et al., 2011; Kermen et al., 2013; Biechl et al., 2017). Olfactory stimulation causes patterns of activity in specific sectors of the olfactory bulb and in the Vv and Dp sectors of the telencephalon in channel catfish and zebrafish (Nikonov, Finger & Caprio, 2005; Yaksi et al., 2009). Blumhagen showed that extracellular recordings of the Dp region of zebrafish are synchronized with the activity of mitral cells after olfactory stimulation (Blumhagen et al., 2011). This is in line with evidence supporting the notion that the Vv and Dp regions of the teleost telencephalon are analogous to structures related to olfactory processing and learning in higher vertebrates (Butler & Hodos, 2005; Ito & Yamamoto, 2009; Mueller et al., 2011).

Olfactory coding in teleosts

The coding of olfactory information regarding stimulus identity, intensity, time course and spatial origin occurs at different levels of the olfactory system, allowing parallel and differentiated processing of different aspects of the olfactory stimuli. Initial coding occurs at the level of the olfactory epithelium and is determined by the types and numbers of ORNs that are activated (Kang & Caprio, 1995; Duchamp-Viret, Duchamp & Chaput, 2000). The information is odotopically organized in the olfactory bulb and modified by efferent input and complex circuits formed by local interneurons, as in higher vertebrates (Li et al., 2019). Finally, a percept is thought to be generated at the level of several telencephalic regions (Wilson & Mainen, 2006). In the olfactory epithelium, ORNs choose and express one allele of the olfactory receptor gene repertoire (Khan, Vaes & Mombaerts, 2011), and receptor neuron groups of variable size that express a common olfactory receptor gene send their axons to the same pair of glomeruli in the bulb. Since odorant receptor proteins are generally of a broad spectrum and activated by different, albeit mostly structurally similar molecules, each odor activates a practically unique combination of ORNs in the olfactory epithelium, and of glomeruli in the olfactory bulb (Malnic et al., 1999; Wilson & Mainen, 2006). The combination of the afferents of many ORNs in the olfactory bulb also causes an increase in signal strength and helps reduce the noise associated with local fluctuations in odorant concentration (Kay & Stopfer, 2006).

In fish, the projections of the diverse types of ORNs reach different sectors of the olfactory bulb, although there is also evidence that a small subpopulation of ORNs projects directly to the ventral region of the telencephalon (Hara & Zielinski, 2007; Tierney, 2015). Electrophysiological studies in trout and channel catfish have confirmed that different types of odorants are processed in different parts of the olfactory bulb, generating a gross general odotopy (Friedrich & Korsching, 1997; Hara & Zhang, 1998; Nikonov & Caprio, 2004; Nikonov & Caprio, 2005; Nikonov, Finger & Caprio, 2005; Rolen & Caprio, 2007). Using retrograde tracers in channel catfish, it was observed that the ciliated ORNs project their axons to medial regions of the olfactory bulb, which are activated by bile salts, and ventral regions, which respond to amino acids, while the microvillous ORNs project mainly to the dorsal surface of the bulb, where mainly responses to amino acids (anterior region) and nucleotides (posterior region) are observed. These separate innervation patterns of the two main types of teleost ORNs were beautifully confirmed in transgenic zebrafish (Sato, Miyasaka & Yoshihara, 2005).

Furthermore, immunohistochemical studies of zebrafish crypt cells have shown that these project to the single posterior mediodorsal glomerulus 2 (mdG2) in the olfactory bulb (Ahuja et al., 2013), which may be involved in pheromone and/or kin recognition (Bazaes & Schmachtenberg, 2012; Biechl et al., 2016). Kappe cells also project to a single glomerulus, posterior mediodorsal glomerulus 5 (mdG5) in zebrafish (Hansen et al., 2003; Hansen et al., 2005; Ahuja et al., 2013; Ahuja et al., 2015; Kress, Biechl & Wullimann, 2014), but their function(s) remain enigmatic to date. Thus, the general odotopic map of vertebrate olfactory bulbs is complemented by a coarser ORN type and odorant class-dependent map in teleost fishes.

Odors usually represent complex mixtures that involve many odorants, and the perception of odors does not generally allow the identification of individual components, unless they are limited in number, as elegantly demonstrated in the catfish Ameiurus melas (Valentincic et al., 2011). Instead, the components are united generating a recognition based on patterns (Laurent, 2002). The olfactory system is able to decompose odors with different degrees of resolution, from a qualitative point of view, thus recognizing the “family” of odorants with which the perceived odor is related. Changes in odor concentration can also be determined, which may affect the perceived smell quality (Friedrich & Laurent, 2001; Laurent, 2002). This process of decoding depends on the complete olfactory system, involving an analysis at the telencephalic level, which acts as a secondary relay of odor information (Yaksi et al., 2009). The first relay, the bulb, performs the task of de-correlating the information, thus increasing the coding space, implying a decrease in the overlap, over time, between odorant mixtures (Friedrich, Laurent & Rainer, 2004). In this process, patterns of oscillatory activity are generated in response to odor stimulation, which are the result of the summation of synaptic activity of multiple afferent neurons (Laberge & Hara, 2001; Friedrich, Habermann & Laurent, 2004; Buzsáki & Draguhn, 2004; Schoppa, 2006). In teleost fishes, olfactory stimulation causes patterns of transient oscillatory activity in specific sectors of the olfactory bulb and in the Vv and Dp sectors of the telencephalon in channel catfish, zebrafish and rainbow trout (Nikonov, Finger & Caprio, 2005; Yaksi et al., 2009; Olivares, 2019) (Fig. 1E). The most prominent oscillations are observed in the 7–12 Hz range, which may functionally correspond to the beta-range in mammals (Satou & Ueda, 1978; Satou, 1990; Kay et al., 2009). As early as in 1938, Adrian & Ludwig, (1938) had been able to observe and record oscillations in the central olfactory system of decapitated fish, using different methods of mechanical and chemical stimulation. Blumhagen et al. (2011), showed that local field potential recordings of the Dp region of zebrafish are synchronized with the activity of mitral cells recorded during olfactory stimulation (Blumhagen et al., 2011). There is mounting evidence indicating that the origin of oscillations in the olfactory system is the olfactory bulb (Satou & Ueda, 1978; Kay, 2014). Although the role of oscillations evoked by odors in the olfactory system of vertebrates is not yet understood, one attractive possibility is that the synchronization of neural networks at specific frequencies allows to isolate transient communication pathways between distant brain regions (Buzsáki & Draguhn, 2004). The different types of oscillations observed in the olfactory system differ in frequency and power depending on the behavioral state of the animal; however, the frequency ranges used for the classification of these signals are species-specific, and generally significantly lower in fishes than in mammals (Kay et al., 2009).

Conclusions

The teleost olfactory system displays important cellular, anatomical and functional differences compared to other classes of vertebrates. These are in part due to the aqueous nature of their habitat, the presence of a sole type of olfactory organ compared to up to five in mammals, and the separation of the olfactory from the respiratory system in fishes. Ciliated and microvillous ORNs have conserved features across vertebrates, whereas olfactory crypt, pear-shaped and Kappe cells appear unique to fishes. The teleost olfactory bulb lacks tufted cells but contains ruffed cells instead, and mitral cells may innervate more than one glomerulus. Olfactory tracts to the telencephalon preserve the bulbar odotopy among their bundles and innervate telencephalic areas that are functionally but not anatomically homologous to higher vertebrates. Prominent field potential oscillations, of lower frequency than in mammals and lacking respiratory rhythms, are observed in response to odorant stimulation, originating in the olfactory bulb and expanding through the olfactory tracts to olfactory telencephalic areas. Future studies will, among other things, define the species-specific odors guiding migratory behavior in certain teleosts, notably in salmonids, clarify how olfactory memories are created and stored, define the roles of pear-shaped and Kappe cells, and hopefully elucidate the function of the olfactory oscillations, which has remained enigmatic for decades.

Additional Information and Declarations

Competing Interests

Author Contributions

Data Availability

The authors declare there are no competing interests.

Jesús Olivares performed the experiments, analyzed the data, prepared figures and/or tables, approved the final draft.

Oliver Schmachtenberg conceived and designed the experiments, analyzed the data, contributed reagents/materials/analysis tools, prepared figures and/or tables, authored or reviewed drafts of the paper, approved the final draft.

The following information was supplied regarding data availability:

This is a review and does not contain any unpublished raw data.

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
