# Peer review of "An update on anatomy and function of the teleost olfactory system"

_PeerJ, doi:10.7717/peerj.7808_

## Round 0.1 · original submission · Minor Revisions

The reviewers have commented on your above paper. They indicated that it is not acceptable for publication in its present form and are recommending various revisions.

If you feel that you can suitably address the reviewers' comments (included), I invite you to revise and resubmit your manuscript.

Reviewer 1 ·

Basic reporting

No Comment

Experimental design

a review: not applicable

Validity of the findings

a review

Additional comments

This review is an update of the excellent 2013 review (J. Chem. Ecol 39:451-464) by the two authors; however , the previous review dealt primarily with fish olfactory receptor neurons, their properties, projections and tuning; the present manuscript includes additional information covering olfactory receptors and higher order olfactory neurons. Of the 100 citations listed in the current manuscript 72% (72 of the 100) were not included in the previous review; however, only 19 of the references cited were published over the past five years (i.e. since the publication in 2013 of the previous review by the present authors). That only 19% of the cited literature provide new information on the topic does not strongly support the necessity of an “update” on the anatomy and function of the teleost olfactory system. The primary message obtained from this “update” is that little new recent information (last five years) has been published on the topic. In spite of this deficiency, the present manuscript along with the suggested alterations will provide a succinct review on the anatomy and function of the olfactory systems in teleosts and be of interest to those who are not specialists in this field.

1. Line(L) 7: single kind? What does this mean? Some fish olfactory organs are arranged into various numbers of lamellae which splay out from a midline raphe, some others have an olfactory organ that contain lamellae but are organized differently (e.g. see Theisen et al. Mar Biol 110:127-135 (1991) Figs 2-4) and still others lack lamellae (Zeiske et al., Cell Tissue Res 172:246-267, 1976).
2. L 88: water flow through fish olfactory organs can be propelled by fish movements, ciliary action of the non-olfactory cilia (in isosmates) and muscular pumping actions (in cyclosmates)-see Doving et al., Acta Zool 58:25-253, 1977 for definitions of cyclosmates and isosmates).
3. L90: true? sniffing is absent (incorrect): see cyclosmates defined in the above Doving reference.
4. L 91-92: Although teleosts lack a separate vomeronasal organ, a recent report (Biechi et al., Scientific Reports 7:44295, 2017) provided evidence for an accessory olfactory system in zebrafish; previous reports in lungfish also indicate the existence of a VNO system (Gonzalez et al., Frontiers in Neuroanatomy 4, article 130, 2010; Nakamuta et al., Anat. Rec. 295:481-491, 2012).
5. L 101-102: There are 4 anatomical types of fish olfactory receptor neurons well-identified, but there is also a fifth that has been reported: “onion-like” (from the Yoshihara lab).
6. L105: “sustain”? Do you mean “contain”?
7. L 150: “middle” should be “”medial”
8. L155:regions indicated (omit “have”); medial olfactory bundle
9. L157: “medial”; omit “would”; relayS
10. Ls 166-179 and throughout the review: Authors seem to like past-perfect tense rather than simple past tense which contains fewer words; e.g. “have been” better as “were”; “has been” better as “was”
11. L 174 Golf: authors need to check the literature as I don’t think this particular G-protein has been demonstrated in fish.
12. L205: (Sato & Suzuki, 2000); however, in seawater….
13. Section titled” “The teleost telencephalon”-especially lines 212-244: The development of the teleost telencephalon is quite complicated with respect to “eversion” which greatly complicates the homologous relationships between telencephalic structures in teleosts vs other vertebrates. A newer review of this situation was published in Neural Development 7, article #212 (2012) that I recommend the authors to review and determine whether their discussion needs updating/alterations especially in light of the authors’ title of this manuscript (e.g. “an update”).
14. The paragraph starting with L 243 contains sentences that are too long; I suggest the following: Omit :although: & start sentence with “The differences..and end the sentence with a period after vertebrates. Start the next sentence with “Studies” and end the sentence with vertebrates. Omit “while” and start the next sentence with Also, the Vs represents….; L 249: substantia innominate, which are related; L 250: omit “on the other hand”-start sentence with “In the dorsal area…; L 251: the pallium, whereas the most medial….end sentence with “amygdala”. L 252: The DI is likely homologous to the hippocampus and the Dp is the main…
15. L 256: omit “allowed” established that the pallial regions and the subpallial nuclei (move all the abbreviations (here) receive projections from the olfactory bulb. Start the next sentence with “The Vv region….
16. L 302: omit “On the other hand”
17. L 307-308: The statement depends on the number of components in a mixture (e.g. see Valentincic et al. J. Fish Biol 79:33-52, 2011).
18. Ls 309—310: the statements concerning “decomposing and “de-correlating” odors” is not well-explained and may not be relevant to the identification of an odorant. This concept is mostly correlated with work presented from the Friedrich & Laurent laboratories as a working hypothesis-however, see Nikonov & Caprio, J. Neurophysiol 98 (2007) p. 1916 “Response time consideration”.
18. L 327 & Figure 1: I recommend some caution when observing recorded olfactory waves especially within the beta-band frequencies and assuming the cause is synchronous neural activity (see Diaz et al. J. Neurosci 27(34): 9238-9245, 2007.

·

Basic reporting

The Review article entitled “An update on anatomy and function of the teleost olfactory
system” by Olivares and Schmachtenberg gives an good but rather short overview on the teleost olfactory system.

The Review article is very well written in a professional English, and it was a pleasure to read it. It is certainly of broad and cross-disciplinary interest and within the scope of the journal.

To my knowledge there are no recent reviews of the teleost olfactory system and I'm happy that the authors made the effort to write this review.

However, I feel that not all the relevant literature has been cited. I strongly encourage the authors to overwork the citations of the manuscript. Every single statement needs an appropriate citation/ reference. (Also see general comments to the authors.)

The unique figure is also not optimal and not very meaningful. I think that some more figures (transduction, general scheme of the fish olfactory system etc.) could substantially improve the manuscript, and make it easier for the potential readers to follow and understand the review.

For the rest, I find myself in the unusual position of not having major complaints. Some minor suggestions can be found in the "General comments to the authors" section.

Experimental design

Not much to complain here.

Validity of the findings

No comment.

Additional comments

Here some minor comments that should be addressed:

-some references that really should be included are:

1: Wanner AA, Genoud C, Friedrich RW. 3-dimensional electron microscopic imaging of the zebrafish olfactory bulb and dense reconstruction of neurons. Sci Data. 2016 Nov 8;3:160100. doi: 10.1038/sdata.2016.100.

2: Wanner AA, Genoud C, Masudi T, Siksou L, Friedrich RW. Dense EM-based reconstruction of the interglomerular projectome in the zebrafish olfactory bulb. Nat Neurosci. 2016 Jun;19(6):816-25. doi: 10.1038/nn.4290.

3: Sato Y, Miyasaka N, Yoshihara Y. Mutually exclusive glomerular innervation by two distinct types of olfactory sensory neurons revealed in transgenic zebrafish. J Neurosci. 2005 May 18;25(20):4889-97.


- In the abstract you speak about 5 types of olfactory receptor neurons. In the relevant part of the review you mention only 4 types. As far as I know, the fifth type of receptor neuron is the pear-shaped type described by Wakisaka et al., 2017. Please correct this incongruity.

- line 34: "....ranging from micromolar to nanomolar....". Here you should could be a little more specific.

- lines 46-48: You distinguish between "social odors" and "pheromonal odors". What is the difference between social- and pheromonal odors? Please clarify or revise this passage.

- lines 64-70: In this passage of your review you should cite also the review of Hamdani el H and Døving KB (The functional organization of the fish olfactory system).

- lines 73-74: What do you mean with "Four older research articles and 22 general articles on vertebrate olfaction were also included in the general data base." . Where do this older and general articles come from?

-around line 110: Here you should consider adding a reference that describes the receptor neurons of terrestrial vertebrates.

-lines 118-131: Here some references are missing. Where did you get the information that you review in this section? Please add some adequate citations.

-line 152: "....chemotropic...." should most probably be "....chemotopic....".

-section "The odor transduction process": Here (or at some other place of thre review) you should mention the adenosine receptor A2c and its transduction cascade described by Wakisaka et al., 2017.

-lines 202-208: It would be great if you could explain a little better what the function of the calcium-activated chloride channel in seawater species is.

- Several times you compare the olfactory system of aquatic and terrestrial vertebrates, but you do not mention the olfactory system of amphibians or other aquatic vertebrates/ tetrapods at all. You should maybe include some information about this.

---

## Round 0.2 · accepted · Accept

Thank you very much for improving your review according to the referee suggestions.